# FULLY CONVOLUTIONAL APPROACH FOR SIMULATING WAVE DYNAMICS

## ABSTRACT

We investigate the performance of fully convolutional networks to predict the motion and interaction of surface waves in open and closed complex geometries. We focus on a U-Net type architecture and assess its ability to capture and extrapolate wave propagation in time as well as the reflection, interference and diffraction of waves. We investigate how well the network generalises both to long-time predictions and to geometric configurations not seen during training. We demonstrate that this neural network is capable of accurately predicting the height distribution of waves on a liquid surface within curved and multi-faceted open and closed geometries, when only simple box and right-angled corner geometries were seen during training. We found that the RMSE of the predictions remained of order $1 \times 10^{-4}$ times the characteristic length of the domain for at least 20 time-steps.

## 1 INTRODUCTION

Predicting the spatio-temporal dynamics of physical systems is a recurrent problem in many areas of science and engineering. A well-established process consists of describing the physical phenomena by human-engineered mathematical models, which capture our current understanding of the physical laws governing the systems, but whose complexity may prevent finding analytical solutions. Scientists therefore frequently turn to numerical solvers to simulate such mathematical models and generate accurate approximations to their solution.

The huge progress in machine learning (ML) algorithms and increased availability of computational power during the last decade has motivated a significant growth in the popularity of data-driven physics. In this field, the interpolation capabilities of neural networks (NNs) have been mostly used in two ways: first, to solve partial differential equations (PDEs) in an unsupervised manner (Dissanayake & Phan-Thien, 1994; Lagaris et al., 1998; 2000; Raissi et al., 2019) and second, to predict the physical dynamics from previous observations without knowledge of the underlying equations (Guo et al., 2016; Farimani et al., 2017; Thuerey et al., 2018; Lee & You, 2019). Unlike the first approach, the latter does not lead to an ana-



Figure 1: Rollouts of our U-Net. It simulates wave motion on a fluid surface with the possible existence of solid walls [video].

lytical representation of the physical dynamics, however, it makes feasible to produce predictions for a diversity of physical domains, boundary conditions and initial conditions without needing to re-train the network, provided that the physical laws are unaltered. Recent studies applying convolutional neural networks (CNNs) to simulate fluid dynamics have reported a speed-up of up to four orders of magnitude, in comparison to traditional numerical solvers, while keeping a similar accuracy (Guo et al., 2016). The major shortcoming of NNs are their often poor generalization to unseen configurations and poor long-time predictions in unsteady simulations.

We investigate the application of fully convolutional neural networks to the problem of forecasting surface wave dynamics, the motion of which is described by the shallow water equations, a system of three non-linear PDEs (Ersoy et al., 2017). Computational modelling of surface waves is widely used in seismology, computer animation and flood modelling (Ersoy et al., 2017; García-Navarro et al., 2019). Our network learnt to simulate a range of physical phenomena including wave propagation, reflection, interference and diffraction at sharp corners. This kind of NN could supplement or potentially replace numerical algorithms used to solve the shallow water PDEs, reducing the inference time by several orders of magnitude and allowing for real-time solutions. This has particular relevance in iterative design scenarios and potential applications such as tsunami prediction.

**Contribution.** We demonstrate that our U-Net architecture is able to accurately predict surface wave dynamics in complex straight-sided and curved geometries, even when trained only on datasets with simple straight-sided boundaries. The generalisation to different initial conditions and longer-time predictions are also evaluated. Additionally, we show how including the MSE of the wave gradient into the loss function significantly reduces spurious oscillations in predicted solutions and may help improve the prediction of the position of the wavefronts. Our network is able to simulate wave dynamics four orders of magnitude faster than a state-of-the-art spectral/$hp$ element numerical solver (Karniadakis & Sherwin, 2013), so it could be an effective replacement for numerical solvers in applications where performance is critical.

## 2 RELATED WORK

**Physics-informed NNs for solving PDEs.** The use of NNs for the solution of PDEs has been investigated since the early 1990s. Most of the relevant research at that time built on the idea of exploiting the universal function approximator property of multi-layer perceptrons (MLPs) (Dissanayake & Phan-Thien, 1994; Dissanayake & Phan-Thien, 1994; Lagaris et al., 1998). In such an approach, the solutions to the PDEs are approximated as MLPs whose only inputs are the spatio-temporal coordinates. These MLPs are trained in an unsupervised way to satisfy the governing PDEs as well as the initial and boundary conditions. The main advantage of this paradigm is that the solution is obtained in a differentiable, closed analytic form, easily usable in any subsequent calculations. Nevertheless, these networks cannot extrapolate to different domains, boundary conditions or initial conditions; making it necessary to re-train the network for every slight modification of the problem. These techniques inspired the more modern physics-informed neural networks (PINNs) (Raissi et al., 2017; Yazdani et al., 2018; Raissi et al., 2019; Lu et al., 2019), which include deeper MLPs and random collocation points.

**CNNs for simulating steady physics.** During the last five years, most of the networks used to predict continuous physics have included convolution layers. For instance, CNNs have been used to solve the Poisson's equation (Tang et al., 2018; Özbay et al., 2019), and to solve the steady Navier-Stokes equations (Guo et al., 2016; Miyanawala & Jaiman, 2018; Yilmaz & German, 2017; Farimani et al., 2017; Thuerey et al., 2018; Zhang et al., 2018). The use of CNNs allows for visual inputs representing physical information, such as the domain geometry or the initial condition, and for visual outputs representing the solution fields. In contrast to PINNs, the network predictions are purely based on observation, without knowledge of the underlying governing equations. This paradigm has proven to extrapolate well to domain geometries, boundary conditions and initial conditions not seen during training (Thuerey et al., 2018). The evaluation of these networks for prediction is considerably faster than traditional PDE solvers, allowing relatively accurate solutions to be predicted between one and four orders of magnitude faster (Guo et al., 2016; Farimani et al., 2017). These reasons make CNNs perfect for developing surrogate models, complementing expensive numerical solvers (Guo et al., 2016; Miyanawala & Jaiman, 2018), or for real-time animations (Kim et al., 2019). Our work takes inspiration from Guo et al. (2016) in the use of a binary geometry field to represent the physical domain. Although Guo et al. (2016); Farimani et al. (2017) and Thuerey et al. (2018) proved the generalisation of their networks to domain geometries not seen during training, these unseen domains contain elementary geometrical entities included within the training data. We go one step further by training the network with exclusively straight boundaries and demonstrating the network is able to generalise to domains incorporating boundaries with varying radius of curvature.

**CNNs for simulating unsteady physics.** Unsteady physics have also been explored from the computer vision perspective (Lee & You, 2019; Sorteberg et al., 2018; Wiewel et al., 2019; Kim et al., 2019; Fotiadis et al., 2020), although to a lesser extent than steady physics. Here, the input to the network is a sequence of past solution fields, while the output is a sequence of predicted solution fields at future times. When predicting unsteady phenomena there is an additional challenge: keeping the predictions accurate along time. To address this, Sorteberg et al. (2018); Wiewel et al. (2019) and Kim et al. (2019) proposed to use encoder-propagator-decoder architectures, whereas Lee & You (2019) and Fotiadis et al. (2020) continued to use encoder-decoder architectures similar to those used for steady problems. Inspired by Fotiadis et al. (2020), which showed that feed-forward networks perform at least as well as recurrent networks in wave forecasting, we opt to use a U-Net architecture (Ronneberger et al., 2015) to perform each time-step of the simulations.

## 3 METHOD

### 3.1 WAVE DYNAMICS DATASETS

The datasets used during training and testing were generated by solving the inviscid, two-dimensional shallow water equations with Nektar++, a high-order spectral/$hp$ element solver (Cantwell et al., 2015). In conservative form, these equations are given by

$$\frac{\partial}{\partial t} \begin{pmatrix} h \\ hu \\ hv \end{pmatrix} + \nabla \cdot \begin{pmatrix} hu & hv \\ hu^2 + gh^2/2 & hvu \\ huv & hv^2 + gh^2/2 \end{pmatrix} = \mathbf{0}, \quad (x,y) \in \mathcal{D} \tag{1}$$

where $g = 9.80665$ m/s$^2$ is the acceleration due to gravity and $\mathcal{D} \in \mathbb{R}^2$ denotes the domain under consideration. Unknown variables in this system are the water depth $h(x, y, t)$ and the components of the two-dimensional velocity vector $u(x, y, t)$ and $v(x, y, t)$.

We imposed two forms of boundary conditions: solid wall boundaries, which result in wave reflection and diffraction; and open boundaries, which allow waves to exit the domain. As initial conditions we considered a *droplet*, represented mathematically by a localized two-dimensional Gaussian superimposed on a unitary depth:

$$h_0^1 = 1 + I \exp\left( -C((x - x_c)^2 + (y - y_c)^2) \right) \tag{2}$$

where $I$ is set to 0.1 m, $C$ is randomly sampled from a uniform distribution between 400 and 1000 m$^{-2}$ and the droplet centre, $(x_c, y_c)$, is randomly sampled from a uniform distribution in $\mathcal{D}$.

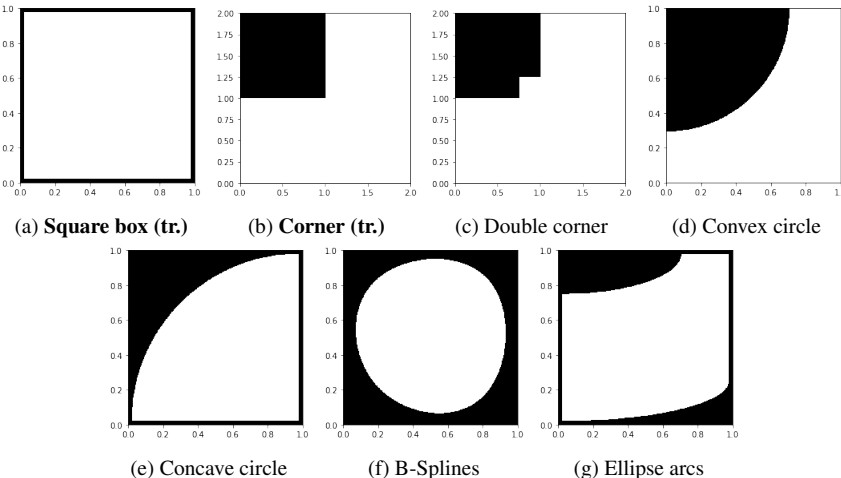

Figure 2: Flow domains on our training (**tr.**) and testing sets. Dimensions in metres.

Each simulation in the datasets is associated with a binary geometry field, $\Omega(x, y)$, which satisfies

$$\Omega(x, y) = \begin{cases} 0, & \text{if } (x, y) \in \mathcal{D}, \\ 1, & \text{otherwise.} \end{cases} \tag{3}$$

Therefore, $\Omega = 0$ inside the fluid domain and $\Omega = 1$ in the solid boundaries (Guo et al., 2016). This geometry forms an additional input to the network, required to provide information about the walls location. Figure 2 shows the geometry field for the seven categories of fluid domains included in the datasets. The table below summarises the training and testing sets. The sequences in each dataset contain 100 snapshots of the height field sampled at intervals of $\Delta t = 0.003$ seconds. For full details of the datasets, see Appendix A. [1]

Table 1: Training and testing datasets

| ID | Dataset | Purpose | Geometry | Initial Condition | Sequences |
|----|---------|---------|----------|-------------------|-----------|
| A | Box_Single_Drop | Training | Figure 2a | Single Drop (eq. (2)) | 500 |
| B | Corner_Single_Drop | Training | Figure 2b | Single Drop (eq. (2)) | 500 |
| C | Steps_Single_Drop | Testing | Figure 2c | Single Drop (eq. (2)) | 200 |
| D | Convex_Single_Drop | Testing | Figure 2d | Single Drop (eq. (2)) | 250 |
| E | Concave_Single_Drop | Testing | Figure 2e | Single Drop (eq. (2)) | 500 |
| F | Spline_Single_Drop | Testing | Figure 2f | Single Drop (eq. (2)) | 200 |
| G | Ellipse_Single_Drop | Testing | Figure 2g | Single Drop (eq. (2)) | 200 |

## 3.2 U-NET AS A SIMULATION ENGINE

Our neural network is based on a U-Net architecture (Ronneberger et al., 2015), which has been extensively used for image-to-image translation tasks (Farimani et al., 2017; Isola et al., 2017; Thuerey et al., 2018; Fotiadis et al., 2020). This architecture consists of a fully convolutional feed-forward encoder-decoder network with skip connections between the encoder and the decoder. In wave dynamics forecasting, the input sequence and the target share an important amount of information at different length scales, this makes the U-Net a particularly appropriate architecture for our problem. Our U-Net receives six fields as input: the geometry field, $\Omega$, and a sequence of five consecutive height fields, $\{h_s, h_{s+1}, h_{s+2}, h_{s+3}, h_{s+4}\}$. It generates as output a prediction of the subsequent height field, $\hat{h}_{s+5}$, at the next time point. Hence, each evaluation of the network corresponds to performing a single time-step, and the network is re-fed with past predictions to make further predictions. See Appendix B for more details about our U-Net architecture.

## 3.3 GRADIENT LOSS

The loss function used in the present work is given by

$$\mathcal{L} = (1 - \lambda)\text{MSE}(\hat{h}_i, h_i) + \lambda\left[\text{MSE}\left(\frac{\partial \hat{h}_i}{\partial x}, \frac{\partial h_i}{\partial x}\right) + \text{MSE}\left(\frac{\partial \hat{h}_i}{\partial y}, \frac{\partial h_i}{\partial y}\right)\right] \qquad (4)$$

where $\lambda$ is a hyper-parameter and MSE is the mean-squared error. The derivatives in equation (4) were computed using second-order finite differences. This loss function penalises oscillations in the predicted fields and favours smooth solutions. This is especially important for temporal forecasting by consecutively re-feeding the network, since the spurious oscillations are amplified in each new prediction. In the case of wave dynamics, the gradients have a large magnitude on the leading and trailing edges of the wavefront. Capturing such small-scale information enables the network to accurately learn and reproduce the wave speed and width of wavefront.

## 3.4 TRAINING

Our U-Net[2] was trained against the simple closed box and open corner geometries shown in Figures 2a and 2b, with a single droplet as initial condition. The time step was set to $\Delta t = 0.012$ seconds and the spatial resolution was set to 128 pix/m. A series of transformations were applied to perform data augmentation and normalisation (see Appendix C). We trained for 500 epochs with the Adam

---

[1] All dataset are available on to be revealed.
[2] GitHub repository available on to be revealed.

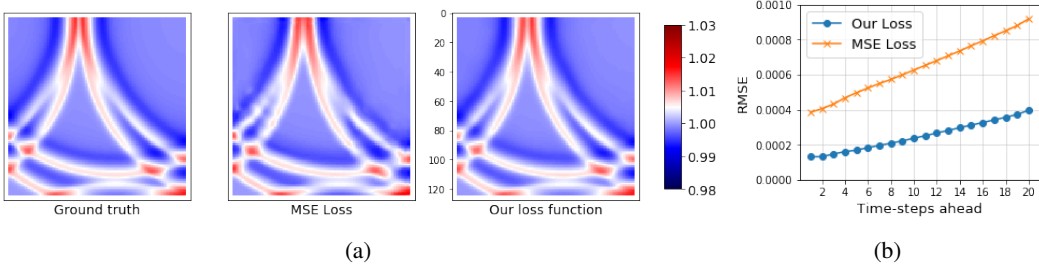

(a)                                                                   (b)

Figure 3: (a) Predictions obtained for the fifth time-step (colour values in metres). (b) RMSE between target and prediction vs. time-step into the future.

optimiser and its standard parameters (Kingma & Ba, 2015) and the loss function given by equation (4) with $\lambda = 0.05$. The learning rate was set to $10^{-4}$ and decreased by a factor of 10 every 100 epochs. The time origin for the input sequences is not at $t = 0$, instead it is randomly selected, and five time-steps are performed before updating the network weights in an auto-regressive fashion.

## 4 RESULTS AND DISCUSSION

In this section we evaluate the ability of the trained model to extrapolate to different domain geometries and initial conditions. We also assess the quality of long-time predictions. [3]

### 4.1 GRADLOSS VS MSE

Figure 3a compares the ground truth and two predictions obtained five time steps ahead of the last temporal height frame used as input. The height field in the central image was predicted by the U-Net trained with the plain MSE, whereas the prediction on the right was obtained after training with the loss function given by equation (4). Here, the suppression of the oscillations on the wavefront when using the GradLoss instead of the MSE is clearly apparent. To quantify this difference, Figure 3b shows the root-mean-squared error (RMSE), averaged over the 500 simulations in dataset A (see Table 1), during 20 time-steps for both loss functions. We can conclude that, even for the first predicted time-step, the network trained with the MSE is considerably less accurate. This difference is amplified linearly with each new predicted frame.

### 4.2 GENERALISATION TO DIFFERENT DOMAIN GEOMETRIES

Our network proved to generalise to complex domain geometries not seen during training, some including curved walls, when only the closed domain and open-corner domain were used for training (Figures 2a and 2b). The three columns of Figure 4b depict the ground truth, prediction and the absolute value of their difference every five time-steps (the input height fields are depicted in Figure 4a). This Figure shows that our network is able to accurately predict the wave speed and correctly infer the reflection on solid walls and diffraction on sharp edges. This is evidenced by the predicted height field at frame 25 (after 20 time-steps), where we observe that the position and wavelength of the predicted and true waves are coincident. The main discrepancy between ground truth and predictions is the height of the wavefronts diffracted on the edges, which are of lower magnitude in the predicted fields. The reason for this may be that the network is less exposed to wave diffraction than to wave propagation and reflection during training.

The previous domain is a straightforward extension of the domain in Figure 2b, which was used during training, so the ability of the network to extrapolate to this configuration is not unexpected. We now explore the generalisation of the network to curved boundaries, which is potentially more challenging. Figures 5a and 5b depict the inputs and outputs for a simulation in the domain depicted in Figure 2d. By visual inspection, these results suggest that the network is able to accurately predict

---

[3]Links to animations comparing the ground truth and the U-Net predictions can be found in https://doi.org/10.6084/m9.figshare.13182623.v1.

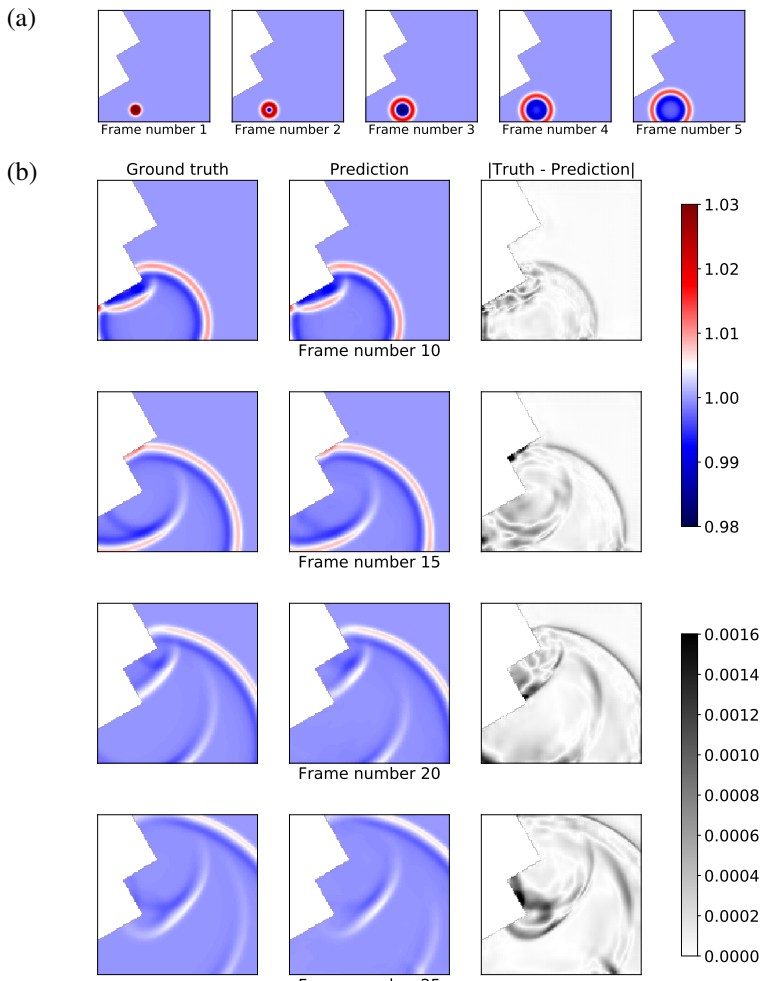

Figure 4: Predictions for a previous unseen partially open geometry, incorporating multiple corners. (a) Input sequence to the network. (b) Ground truth (left column), predictions (centre) and absolute difference (right) for such inputs. Colour values in metres [video].

wave reflections on circular convex walls, and errors are of the same magnitude to those in Figure 4b. The same conclusion was obtained for circular concave walls.

The network is also able to produce good-quality predictions for configurations involving reflections on walls whose radii of curvature is not uniform, although in these cases more substantial differences between targets and predictions can be seen. The temporal frames depicted in Figure 1 illustrate the good generalisation to a fluid domain surrounded by spline-sided walls. The successful generalisation to this kind of domain is likely due to the localised support of the convolution kernels and the architecture. When the convolutions are applied, the network may interpret curved walls as polygonal walls made up of lots of many straight walls of size equal to the stride size. In that case, the higher the image resolution, the smaller the radius of curvature that could be handled by the network.

### 4.3 Generalisation to a Higher Number of Droplets

Our U-Net demonstrated a satisfying accuracy on simulations with two or more droplets in the initial condition. For instance, Figure 6 shows the solution field after 20 time-steps on a domain surrounded by curved walls. Here, the resemblance between the ground truth and the network prediction evidences the proper propagation and reflection of the initial excitations while they interfere with each

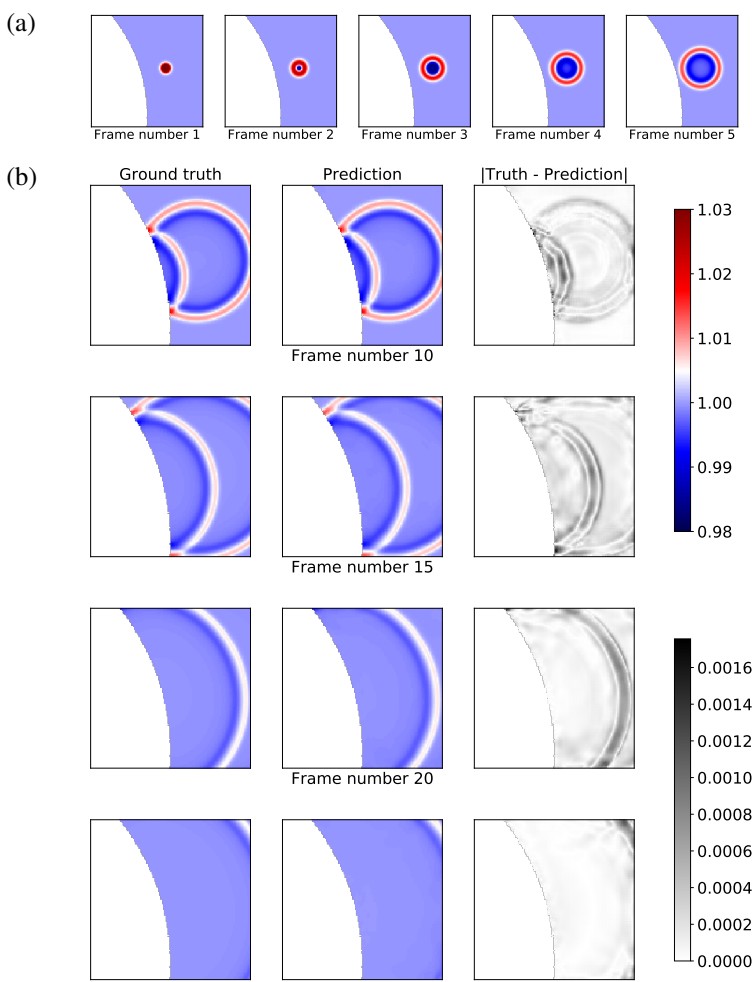

Figure 5: Prediction for a previously unseen partially open geometry, incorporating a curved boundary. (a) Input to the network. (b) Ground truth (left column), predictions (centre) and absolute difference (right) for such inputs. Values in metres [video].

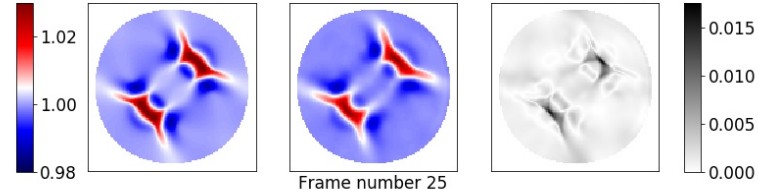

Figure 6: Ground truth (left column), predictions (centre) and absolute difference (right) after 20 time-step and two initial droplets. Values in metres [video].

other. The height field depicted in Figure 7 was obtained predicting the evolution of four droplets for 20 time-steps. While this prediction is far from exact, we can observe that the network reproduces well the complex height patterns originated in this chaotic simulation.

## 4.4 LONG-TIME PREDICTIONS

Firstly, we would like to highlight that the size of the time-steps performed by our network are considerably bigger than in previous work. For instance compared to the work of Fotiadis et al.

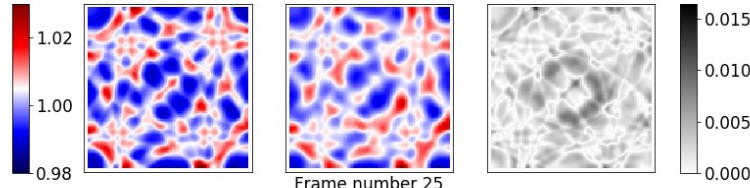

Figure 7: Ground truth (left column), predictions (centre) and absolute difference (right) after 20 time-step and four initial droplets. Values in metres [video].

(2020), our ratio $\Delta t/t_c$ is four times bigger (we consider the characteristic time scale of the shallow water equation defined as $t_c = \sqrt{L/g}$). This challenges the network accuracy, but also decreases the number of time steps required to simulate certain time interval, which decreases the computational time. This is another advantage of NN-based solvers, since traditional PDE solvers must keep a low time-step size due to stability constraints.

We assessed the ability of the network to make accurate predictions over longer time periods by simulating in a domain four times longer. Figure 8 shows predictions and ground truth after 30, 55 and 80 time-steps. The initial condition is a single droplet placed in the centre of the narrow channel. The network is able to predict accurately the speed of the leading wavefront, however, the wave magnitude is not retained and it is not able to maintain the propagation of the wavefronts originating from reflections within the narrower channel, which can be seen to quickly dissipate. This problem could be addressed by training the U-Net for longer output sequences and possibly increasing the depth of the network.

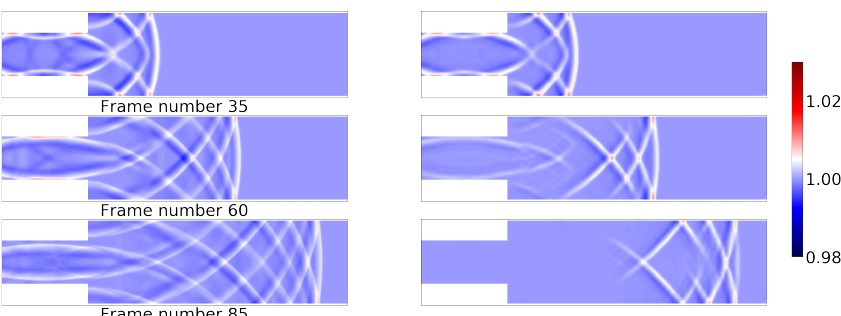

Figure 8: Ground truth (left) and predictions (right) for simulations in a channel with a sudden expansion. The initial condition is a droplet placed at the middle of the narrower segment. Values in metres.

## 4.5 BASELINE

We contrast now our U-Net with the one proposed by Fotiadis et al. (2020). In their work, they compared a U-Net, a LSTM, a ConvLSTM and a Casual LSTM for the task of forecasting surface waves; and they demonstrated the superiority of the U-Net in achieving high accuracy and low computational time. Our U-Net has a smaller depth than the one in Fotiadis et al. (2020), resulting in 4.2 times less learnable parameters and faster network evaluations. However, each forward pass of their U-Net returns predictions for the next 20 time-points, instead of the single time-step performed by our network. As a result our simulations are one order of magnitude slower, but also about one order of magnitude more accurate. We trained the U-Net in Fotiadis et al. (2020) in a similar way to ours (see section 3.4), but with the MSE as loss function and predicting 20 frames as they did in their original work. Figure 9 shows the RMSE between the ground truth and these two U-Nets predictions for our datasets in Table 1. We can appreciate that the RMSE of our U-Net predictions increases linearly with time, whereas the RMSE of the U-Net in Fotiadis et al. (2020) remains approximately constant after the $5^{th}$ time-point, but considerably higher. The decrease in the RMSE in the datasets B, C and D is explained by the wavefronts leaving the fluid domain due to the presence of open

boundaries. From these results we can conclude that the network depth in Fotiadis et al. (2020) is unnecessarily big, and more important, that training in an auto-regressive fashion for five time-steps is better than training for 20 frames predicted in a single evaluation of the network.

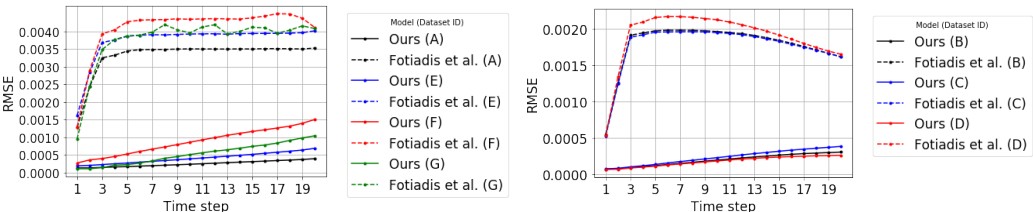

Figure 9: RMSE between target and prediction for the datasets in Table 1.

## 4.6 PERFORMANCE

Performing one time step with our network is 62 times faster than simulating that same time-interval in Nektar++ when the same single-core CPU is used. On the other hand, performing the network evaluation on a Tesla T4 results in a 21765x speed up with respect to Nektar++ running on the CPU. This improvement, and the low RMSE reported in the different datasets, should be enough to justify the use of our model for fast-solution applications.

## 5 CONCLUSION

In this work, we investigated the application of fully convolutional deep neural networks for forecasting wave dynamics on fluid surfaces. In particular, we focused on a U-Net architecture with two skip connections, and we trained the network to predict the spatio-temporal evolution of wave dynamics, including: wave propagation, wave interference, wave reflection and wave refraction. We demonstrated that including the MSE between the gradients of the predicted height and the truth height in the loss function significantly reduces spurious oscillations in the solution and helps to predict the position of the wavefronts more accurately. This suggests that loss functions which capture discrepancies in the predicted spatial gradients provide valuable information when training networks to forecast wave dynamics, especially over long-time intervals. The domains considered during training only included a closed box and a single right-angled corner. However, our U-Net was able to extrapolate to curved walls with varying radii of curvature. The RMSE remained of order $10^{-4}$ times the characteristic length for at least 20 time steps for both the training and testing datasets. When run on a GPU, these simulations are around $10^4$ times faster than the equivalent numerical simulation used for generating our datasets. These findings highlight the potential for neural networks to accurately approximate the evolution of wave dynamics with computational times several orders of magnitude smaller than conventional numerical simulation.

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

## A  DATASETS

Datasets A-C contain only straight-sided fluid domains. Dataset A only includes wall boundary conditions, whereas B and C include open boundaries. Datasets D-G contain curved-sided domains. The domains in dataset F were generated randomly given four random control points and using B-Splines to create a closed domain. The domains in dataset G were also generated randomly given a concave quarter of an ellipse and a convex quarter of an ellipse, whose minor axes follow a uniform distribution between 0.25 and 0.5 m. Only the datasets A and B were used during training. The remaining datasets were used to demonstrate the ability of the network to generalise to unseen fluid domain geometries.

## B  U-NET ARCHITECTURE

The diagram in Figure 10 depicts the U-Net architecture used in the present work. The network receives six fields as input: the geometry field $\Omega$ and a sequence of five consecutive height fields $\{h_s, h_{s+1}, h_{s+2}, h_{s+3}, h_{s+4}\}$. The output is a prediction of the subsequent height field $\hat{h}_{s+5}$. Whereas some recent studies (Fotiadis et al., 2020; Thuerey et al., 2018) have used bi-linear interpolation to perform the up-sampling, we opted to use transpose convolutions with a 2x2 kernel and stride 2. This increases the number of trainable parameters to 1,864,577, but we also noticed a significant improvement in the quality of the predictions.

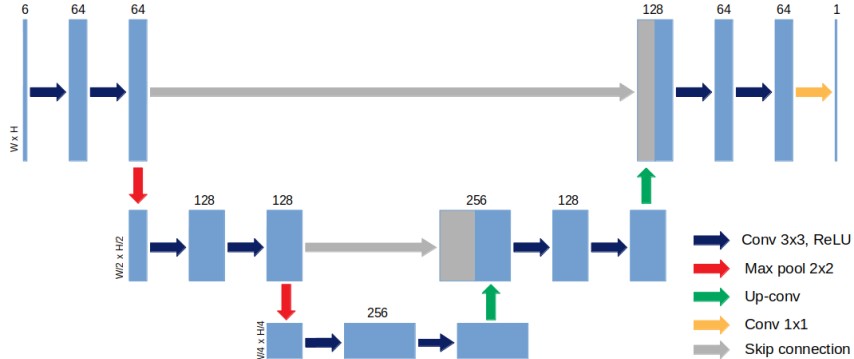

Figure 10: Our U-net architecture with 1,864,577 trainable parameters. It receives six fields as input: the geometry field and a sequence of five consecutive height fields. The output is a prediction of the height field in the subsequent time step.

## C  NORMALISATION AND DATA AUGMENTATION

To improve generalisation across a range of wave dynamics, the height fields were re-scaled according to $\tilde{h} \leftarrow (h - \bar{h})/(\max(h) - \bar{h})$, where $\bar{h} = 1$ and $\max(h) = 1.1$. This re-scaling was reversed for visualising the network predictions. In order to avoid over-fitting and improve the generalisation capabilities of the network, we apply two sets of transformations to the training datasets. For the dataset $A$ such transformations consist on random rotations of 90, 180 and 270 deg as well as horizontal and vertical flips. For the dataset $B$ random rotations by multiples of 15 deg are applied and the physical size of the frames is reduced to 1 m $\times$ 1 m by cropping the original frames to domains whose center position follows an uniform distribution from 0.9 to 1.1 m in both spatial directions. Finally, the images of all these sequences were linearly interpolated to a 128 $\times$ 128 resolution to satisfy the 128 pix/m requirement. Regarding the testing datasets, sets $E$, $F$ and $G$ were augmented in the same manner as the dataset $A$, since they contain only closed boundaries; and datasets $C$ and $D$ are augmented like $B$, as they include open boundaries.

