# OpenReview forum: "Fully Convolutional Approach for Simulating Wave Dynamics"
_ICLR.cc/2021/Conference — Reject_

### Official Review · AnonReviewer1 · 2020-10-27
**Interesting idea, but needs more experiments**

**Rating:** 5
**Confidence:** 4

**Review:**

The authors use a U-Net architecture network to predict the motion and interaction of surface waves in an open and closed complex. The network trained on data with a simple box and right-angled corner geometries and generalizes well to other complex geometric configurations. The neural network-based method runs much faster than the standard numerical simulation by directly solving the PDE.

Strengths:
1. Much faster than directly solving the inviscid, two-dimensional shallow water equations using standard numerical solver.
2. The network can generalize to geometric configurations and initial conditions not seen during training.

Weaknesses:
1. The predictions over more extended periods are not very accurate. Although training the U-Net for longer output sequences and increasing the network's depth may help, this may also increase the difficulty in training.
2. Some previous works also used the U-net to predict wave dynamics as [3]. It is not clear what is the novelty (if any) in the proposed network architecture.
3. Not enough Experiments. How does the model generalize with more complicated initial conditions, for example, five or ten droplets?  Furthermore, there is no comparison to other existing work.

A further question:
The model generalizes well to other boundary geometries. Is this because the wave reflection at the boundary is a local phenomenon, so the convolutional network only needs to learn the local reflection?


Some relative works:
[1] Zhu, Weiqiang, Yixiao Sheng, and Yi Sun. "Wave-dynamics simulation using deep neural networks." (2017).
[2] Sorteberg, Wilhelm E., et al. "Approximating the solution to wave propagation using deep neural networks." arXiv preprint arXiv:1812.01609 (2018).
[3] Fotiadis, Stathi, et al. "Comparing recurrent and convolutional neural networks for predicting wave propagation." arXiv preprint arXiv:2002.08981 (2020).

---

### Official Review · AnonReviewer4 · 2020-10-28
**Ok submission, but needs more novel model or application domain (e.g with longer time sequences)**

**Rating:** 4
**Confidence:** 4

**Review:**

Summary: The paper applies a fully convolutional U-Net model for next step prediction of the height field for 2d wave dynamics. On the domains tested, the predictions remain accurate for 20 time-steps and the method seems to provide considerable speed-ups compared to a state-of-the-art spectral/hp element numerical solver. Some encouraging results on generalization to new domain shapes and larger scale domains are also presenting in the experimental session.


Strong points:
The manuscript is very well written, the methods are clear and the generalization results are encouraging.

Weakness:
1) The paper lacks a novel contribution from the architectural and application side: UNets have been previously used on forward dynamics predictions (e.g. [1] Thuerey et al.) and there are other works using neural networks on wave prediction (e.g. [2] Fotiadis et al.). The architecture is a straightforward application of a Unet architecture + a loss function with MSRE of the predictions and its gradients.

2) There are no comparisons to other baselines. The code for the UNet architecture from [1] Thuerey et al. is open sourced, so this would be a natural baseline to include.

3) The MSRE errors are not reported on the full dataset, instead plots for single trajectories are presented.

4) Many of the modeling choices are not well motivated or explained, for example

a) the training set dt=0.03s and for unrolling it's set dt=0.12s, but no insight on this choice was provided.
b) a history of the previous 5 time-steps is used as an input to the network, no ablation or justifications are provided.
c) in the training details, authors state "The time origin for the input sequences is not at t = 0, instead it is randomly selected, and five time-steps are performed before updating the network weights" but nothing else is discussed.
d) authors claim that their model is 4 order of magnitude faster than a state-of-the-art spectral/hp element numerical solver, but the actual runtimes per dataset are not provided, and there is no discussion of terms of scalability, e.g, how does it scale with the space resolution.
e) In the generalization to a large domain experiment (section 4.4), the authors justify the relatively poor performance of the approach by stating "This problem could be addressed by training the U-Net for longer output sequences and possibly increasing the depth of the network." Since this is a simple hypothesis, I feel it should have been tested.

5) Most experiments are showing results for 20 time-steps unrolls, and while for longer sequences dissipation was observed -- which is a strong limitation of the model.


Because of the points above, I think this manuscript does not pass the ICLR threshold for acceptance, although I believe this would be a good workshop paper submission.

[1] Nils Thuerey, Konstantin Weissenow, Lukas Prantl, and Xiangyu Hu. Deep Learning Methods for Reynolds-Averaged Navier-Stokes Simulations of Airfoil Flows
[2] Stathi Fotiadis, Eduardo Pignatelli, Mario Lino Valencia, Chris Cantwell, Amos Storkey, and Anil A. Bharath. Comparing recurrent and convolutional neural networks for predicting wave propagation

---

### Official Review · AnonReviewer3 · 2020-10-30
**Solid experimental paper**

**Rating:** 7
**Confidence:** 4

**Review:**

In this paper, a methodology for simulating wave dynamics is presented based on convolutional neural networks. A standard analytic wave dynamics solver was used to generate a large dataset of 2D wave simulations. A deep net based on U-Net was trained to predict the next state of the wave field given the five previous states. The training was done with both standard MSE loss and also GradLoss, which placed a loss on the gradients of the field as well. The results show that the network trained with GradLoss performs significantly better over successive rollouts than the one trained with MSE. The results also show that the trained network is able to generalize to qualitatively different environments from the training set.

If there was one word I would use to describe this paper, it is thorough. The introduction and related work do a good job going over related contributions in the literature and giving the reader, particularly a reader with an ML background and not necessarily a fluid dynamics background, a good understanding of what has been done to this point. The same can be said for the results. The paper offers thorough analysis of the technique. The results first show why GradLoss is necessary. They then go on to show the performance of the model in a variety of scenarios qualitatively different than the scenarios seen during training, including curved surfaces and multiple droplets. The model performs well in these scenarios. Additionally, the paper presents a scenario that the model fails at (long term predictions with two droplets in a narrow channel) and briefly discusses why the model fails. The strength of this paper is in the thoroughness of its results, which includes presenting results where the model fails.

The weakness of this paper is that it doesn't present any novel techniques. It's an existing architecture (U-Net) applied in a new domain (wave simulation). However, experimental papers provide value as well, and the field is grossly lacking in them. To that end, this paper could be a useful contribution. The results are thorough and even show the limitations of the model. For a full experimental paper, it would be nice to do even more analysis, although with limited number of pages that may not always be possible. However I would have liked to see have seen data showing how much faster this model is over the standard solvers. The paper claims in the intro and conclusion that their model is 10^4 times faster than the standard method, but no where in the paper is data provided to back this up. This seems like one of the main reasons for using this methodology, so not having data to show this is a big oversight. Still overall, while there is room for improvement, this is a quality paper and should be accepted.

A couple small things to fix up for the camera ready version:
-A related paper that should probably be cited "Accelerating Eulerian Fluid Simulation With Convolutional Networks" that does very similar things with deep nets and fluid dynamics.
-Fig 7 says predictions are on the left and ground truth is on the right, but I believe that is flipped (predictions are on the right, ground truth is on the left).

---

> ### Comment · Area_Chair1 · 2020-11-16
> **thoughts on experimental issues highlighted by R4?**
>
> Thanks for your review!
>
> What do you think of the potential experimental issues that AnonReviewer4 raised?

---

> > ### Comment · AnonReviewer3 · 2020-11-17
> > **Thoughts**
> >
> > Reviewer 4 brings up some fair points. I'm not as concerned about the lack of a baseline comparison; that doesn't seem to be the point of this paper, although having one can't hurt. Not reporting MSRE on the full dataset is an issue that I missed that reviewer 4 caught, and I think that it is actually important. It seems small enough that the authors could add it for the final version of the paper, although I'm hesitant to suggest that route because papers should be largely final when reviewed. The issue about rollouts only being for 20 timesteps is a good one, and it would be nice to see longer rollouts. I would suspect with such a chaotic system, longer rollouts would probably diverge quite a bit.
> >
> > Overall, while I support reviewer 4's call for more experiments, there is only so much that can be done in an 8 page conference paper. Page limits really force authors to strip out a lot, and I'm really not a fan of the recent trend of putting more and more "details" in the appendix. Even in light of the other reviewers's comments though, I stand by my decision on this paper. However, given that the other 2 reviewers think the paper could use more work, it would be completely reasonable for the chairs to reject it based on those reviews. I think it's fine as is, but I'm okay with the chairs disagreeing.

---

> > > ### Author Response · Authors · 2020-11-18
> > > **Fixed on issues**
> > >
> > > Thank you for your reviews, they are being helpful in improving how our work is delivered. Here are some of our thoughts and improvements we will add to the final version:
> > >
> > > 1. The objective of the paper was not to compare the results with previous networks, but to demonstrate the generalisation to geometries completely different to the ones seen during training, a topic relevant in physical simulations and that has not been addressed to this extent by previous authors studying wave dynamics. However, we are going to include a comparison with the U-Net used in Fotiadis et al. That U-Net has 4x times more weights and is trained to predict 20 frames; whereas ours is trained to predict 5 frames,  one a time in an autoregressive fashion.
> > >
> > > 2. We will include the RMSE of the whole datasets in an appendix, we did not include before due to the lack of space.
> > >
> > > 3. The network producing rollouts of length 20 is not really an issue, but a consequence of having a bigger time-step size between frames (which is good for getting fast simulations). When we want to run a simulation using traditional solvers we are constrained to use small time-step sizes (due to stability and accuracy issues), however, neural networks allow for much bigger time-steps. Bigger time-step size means fewer time steps to be performed during a simulation, and hence lower computational time. So for us, it is not so important the length of the rollouts, but the ratio (period of physical time simulated)/(computational time of the simulation).
> > >
> > > 4. We will include experiments with 2-4 droplets in the initial condition.
> > >
> > > P.D. These modifications have been already added in the reviewed submission. The previous points can be found now in the following sections:
> > >
> > > 1- 4.5 Baseline
> > >
> > > 2- 4.5 Baseline
> > >
> > > 3- 4.4 Long-time predictions
> > >
> > > 4- 4.3 Generalisation to a higher number of droplets

---

### Official Review · AnonReviewer2 · 2020-11-03
**Lack of novelty; not of publishable quality**

**Rating:** 3
**Confidence:** 4

**Review:**

### Summary and Contributions
The paper proposes a generative model for the motion of surface waves in open and closed geometries. While neural networks have been applied to simulate fluid dynamics before, they are purported to suffer from poor generalization to unseen geometries for long-time predictions. The paper proposes a U-net based model which is trained using a modified loss function incorporating the gradient into the loss function.  The results demonstrate generalization to unseen geometries with good predictions upto 80 time steps in the future.

### Detailed Review
The following is the detailed review of the paper, organized into strengths and weaknesses subsections.

### Strengths

#### Relevance and Significance
The utilization of DNNs for modeling physical phenomena is compelling and gaining steam. In particular, the modeling of spatiotemporal phenomena like unsteady fluid dynamics for complex geometries should be of interest to the ML community.

#### Clarity
The paper Is written well and is easy to understand.

#### Reproducibility
Should be reproducible.

### Weaknesses
#### Relation to Prior Art
The paper does a reasonable job of presenting the prior art but fails at identifying the unmet need that the presented work fulfills. The claimed speedup over numerical solvers has been achieved before (Guo et al, 2016). Further, DNNs have been used to model surface waves (Fotiadis et al, ICLR Workshop 2020, Sorteberg et al, NeurIPS Workshops, 2018, Kim et al Eurographics 2019,  etc.). (Sorteberg 2018) claim to use LSTM for modeling up to 80 time steps into the future on a dataset not seen during training. It is not clear what advantages the present work is supposed to have over the SOTA and why.

#### Methodology
The proposed approach is a straightforward application of U-net to predict a spatial field given past few spatial fields (stacked together). However, U-Nets, LSTMs, conv-LSTMs and other architectures have been tried before. It is unclear what the novel contribution in this paper is (gradient augmented loss function?) and why it would be instrumental in handling unseen geometries over longer periods of time.

#### Novelty
The presented approach is a straight-forward application of known techniques. Further, these approaches have been tried before (see prior art)

#### Empirical Evaluation
There is no evaluation against the state of the art. It is important to compare the performance against (Fotiadis et al, ICLR 2020 Workshops),  (Sorteberg et al, NeurIPS Workshops, 2018), (Kim et al Eurographics 2019),  etc.

### Assessment
Though the problem seems relevant and of significance to the research community, the paper suffers from a lack of novelty and a non-existent comparison against the state of the art. It fails to identify the unmet needs it is addressing and what novel contributions allows them to achieve this. I do not recommend the publication of this paper.

---

### Decision · Program_Chairs · 2021-01-07
**Final Decision**

**Decision:**

Reject

**Comment:**

Reviews were somewhat mixed here, but the consensus is to reject, with at least one voice (R2) urging rejection. Across reviewers, the recommendation to reject is primarily based on the level of originality with the proposed U-Net architecture and on weakness of experiments, especially in comparing to baselines.

Reviewers found strengths in the paper's writing and in its demonstration of generalization to unseen geometries.

However, reviewers noted that the architecture does not win originality/significance points (including R3, the most positive reviewer):
* R3: "The weakness of this paper is that it doesn't present any novel techniques. It's an existing architecture (U-Net) applied in a new domain (wave simulation)."
* R2: "The proposed approach is a straightforward application of U-net to predict a spatial field given past few spatial fields (stacked together). However, U-Nets, LSTMs, conv-LSTMs and other architectures have been tried before. It is unclear what the novel contribution in this paper is [...] and why it would be instrumental in handling unseen geometries over longer periods of time."
* R2 post-response: "This paper is a clear reject. None of the contributions are novel [...]"
* R4: "The paper lacks a novel contribution from the architectural and application side"
* R1: "Some previous works also used the U-net to predict wave dynamics [...] It is not clear what is the novelty (if any) in the proposed network architecture"

Reviewers also noted weaknesses in the experiments (acknowledged by R3, the most positive reviewer, though that review did not consider them a fatal flaw):
* R1: "Not enough Experiments. How does the model generalize with more complicated initial conditions, for example, five or ten droplets? Furthermore, there is no comparison to other existing work."
* R2: "There is no evaluation against the state of the art [...]"
* R2 post-response: "Application of DNNs to this problem, speed-ups over numerical solvers, etc. have all been explored by SOTA works which have not been compared against. There is no clear articulation of the claimed novel contributions over the SOTA and empirical validation (or theoretical reasoning) of the same."
* R4: "There are no comparisons to other baselines [...] "
* R3: "Reviewer 4 brings up some fair points [about experimental issues]. I'm not as concerned about the lack of a baseline comparison; that doesn't seem to be the point of this paper [...] there is only so much that can be done in an 8 page conference paper [...] However, given that the other 2 reviewers think the paper could use more work, it would be completely reasonable for the chairs to reject it based on those reviews."

Based on this consensus of reviews, my recommendation is to reject. I hope the feedback from the reviews is helpful to the authors.